# DNA Vaccines Encoding HTNV GP-Derived Th Epitopes Benefited from a LAMP-Targeting Strategy and Established Cellular Immunoprotection

**DOI:** 10.3390/vaccines12080928

**Published:** 2024-08-19

**Authors:** Dongbo Jiang, Junqi Zhang, Wenyang Shen, Yubo Sun, Zhenjie Wang, Jiawei Wang, Jinpeng Zhang, Guanwen Zhang, Gefei Zhang, Yueyue Wang, Sirui Cai, Jiaxing Zhang, Yongkai Wang, Ruibo Liu, Tianyuan Bai, Yuanjie Sun, Shuya Yang, Zilu Ma, Zhikui Li, Jijin Li, Chenjin Ma, Linfeng Cheng, Baozeng Sun, Kun Yang

**Affiliations:** 1Department of Immunology, The Key Laboratory of Bio-Hazard Damage and Prevention Medicine, Basic Medicine School, Air-Force Medical University (the Fourth Military Medical University), Xi’an 710032, China; superjames1991@foxmail.com (D.J.); zjq000211@163.com (J.Z.); 18703383373@139.com (W.S.); sunyubo000103@163.com (Y.S.); wzj031129@163.com (Z.W.); vv_jevin@163.com (J.W.); jinpeng_zhang@foxmail.com (J.Z.); wengweng24@foxmail.com (G.Z.); kui_bei@foxmail.com (G.Z.); wangyy_666@163.com (Y.W.); 13572472008@163.com (S.C.); jiaxingzhang09@163.com (J.Z.); wangyongkaiyx@163.com (Y.W.); 18391800515@163.com (R.L.); tianyuanbai@163.com (T.B.); syjfly@163.com (Y.S.); yangshuxiaoya@163.com (S.Y.); 13930296799@163.com (Z.M.); lizhikui1998@163.com (Z.L.); qjmj856915ljj@163.com (J.L.); mcj1359364040@163.com (C.M.); 2Department of Microbiology, Basic Medicine School, Air-Force Medical University (the Fourth Military Medical University), Xi’an 710032, China; chenglfz@fmmu.edu.cn; 3Yingtan Detachment, Jiangxi General Hospital, Chinese People’s Armed Police Force, Nanchang 330001, China

**Keywords:** Hantaan virus (HTNV), glycoprotein (GP), Th epitopes, recombinant antigen, LAMP-targeting

## Abstract

Vaccines has long been the focus of antiviral immunotherapy research. Viral epitopes are thought to be useful biomarkers for immunotherapy (both antibody-based and cellular). In this study, we designed a novel vaccine molecule, the Hantaan virus (HTNV) glycoprotein (GP) tandem Th epitope molecule (named the Gnc molecule), in silico. Subsequently, computer analysis was used to conduct a comprehensive and in-depth study of the various properties of the molecule and its effects as a vaccine molecule in the body. The Gnc molecule was designed for DNA vaccines and optimized with a lysosomal-targeting membrane protein (LAMP) strategy. The effects of GP-derived Th epitopes and multiepitope vaccines were initially verified in animals. Our research has resulted in the design of two vaccines based on effective antiviral immune targets. The effectiveness of molecular therapies has also been preliminarily demonstrated in silico and in laboratory animals, which lays a foundation for the application of a vaccines strategy in the field of antivirals.

## 1. Introduction

*Orthohantavirus hantanense*, belonging to the order Bunyavirus, family Hantaviridae, and genus Orthohantavirus, are responsible for two distinct zoonotic diseases in humans: hemorrhagic fever with renal syndrome (HFRS) [1] and hantavirus cardiopulmonary syndrome (HCPS) [2]. HFRS is primarily found in Eurasia and is usually characterized by mild to moderate symptoms, including fever, headache, and gastrointestinal issues, which can progress to hypotension and acute renal failure; however, it has a relatively low fatality rate of 1% to 15% [3]. In contrast, HCPS occurs in the Americas and presents with non-cardiogenic pulmonary edema and respiratory distress, lacking the renal complications seen in HFRS, with a significantly higher fatality rate of up to 60% [4].

HFRS is caused mainly by the Hantaan virus (HTNV). It has three negative-sense, single-stranded RNA segments, large (L, 6456 bp), medium (M, 3408 bp), and small (S, 1290 bp), which encode an RNA-dependent polymerase (2151 aa), an envelope glycoprotein (GP, N-terminal Gn and C-terminal Gc, 1135 aa), and a nucleocapsid protein (NP, 429 aa) [5]. The glycoproteins Gn and Gc form heterodimers that assemble into tetrameric spikes, facilitating viral entry via receptor-mediated endocytosis and endosomal membrane fusion, making them key targets for neutralizing antibodies [6,7]. The prevalence of hantaviral diseases is mainly associated with the cross-species transmission of rodents [8,9] and climate change in both autumn and spring [10,11]. The pathophysiology involves the disruption of the endothelial cell barrier, increased capillary permeability, thrombocytopenia due to platelet activation and depletion, and an exaggerated immune response [12]. Symptoms in patients are clinically characterized by the acute onset of fever, headache, abdominal discomfort, acute kidney injury, and hemorrhaging [13].

Due to the lack of effective antiviral drugs [14] and inactivated vaccines, without global ratification, there is still a public health protection barrier in natural epidemic areas [15]. In view of viral biohazards, the development of new preventive vaccines has attracted the attention of biosafety research in both developed and developing countries [16,17]. Over the past decade, USAMRIID has established viral GP-based DNA candidate vaccines against multiple strains of HTV and has made many attempts at delivery [18,19]. Protective T-cell epitopes play a crucial role in the immune response to infections and are essential for the development of effective vaccines. When a pathogen enters the body, specific epitopes are presented to them by antigen-presenting cells. This triggers the T-cells to produce cytokines and other molecules that help to eliminate the pathogen. In vaccine development, identifying and including protective T-cell epitopes in the vaccine formulation can enhance the immune responses and provide long-lasting protection against the pathogen [16,20]. Previous studies have shown that LAMP can anchor the lysosomal membrane and be transported to the MHC II compartment (MIIC). The lysosome-associated membrane protein 1 (LAMP 1) targeting strategy has been widely researched and has shown great promise in vaccine development [21,22,23].

Protective T-cell epitopes and LAMP1-targeting practices are two major academic contributions to the exploration of HFRS immunity and prevention [24,25,26]. In this study, based on the reasonable design of a recombinant antigen that covers HTNV GP-derived protective Th epitopes, novel DNA vaccines were constructed and evaluated, and an effective antiviral protective immune response was established with the assistance of LAMP1, providing a theoretical and practical basis for the development of antiviral therapy involving epitope vaccines.

## 2. Materials and Methods

### 2.1. Ethics Statement

This study involved animal research conducted in compliance with the “Regulations for Operation of Laboratory Animals” established by the Laboratory Animal Centre of the Air Force Medical University in Xi’an, China. The Animal Ethics Committee at the Air Force Medical University approved the animal care and use procedures (No. FMMU-DWZX-20221201). All experiments involving biohazardous materials, including live viruses and infected cells and animals, were performed in a laboratory adhering to biosafety level 3 (BSL-3) standards.

### 2.2. Viruses, Bacteria, Cells, the Inactivated Vaccine, and Peptides

The HTNV strain 76-118 was grown in BHK-21 cells, while *Escherichia coli* (*E. coli*) DH5α and HeLa cells were stored at −80 °C. These cells were cultured in Roswell Park Memorial Institute (RPMI)-1640 medium supplemented with 10% fetal bovine serum at 37 °C with 5% CO_2_.

The inactivated bivalent HFRS vaccine (HANPUWEI^®^) used in this research was produced by the Changchun Institute of Biological Products Co., Ltd. in Changchun, China. The vaccine contains inactivated and purified HTNV and SEOV, cultivated in hamster kidney cells, and is the only bivalent HFRS vaccine available in China.

The 51 highly immunoreactive peptides involved in the synthesis of HTNV GP in Gnc were cloned and inserted into a peptide synthetic from the company ChinaPeptides (Shanghai, China), and only 47 peptides could be obtained because of the inherent difficulties in their production. The single peptides were diluted in PBS at a concentration of 20 μg/mL for the ELISpot assays (Appendix A).

### 2.3. Construction of Recombinant DNA Vaccines

The proteinic Gnc sequence of the HTNV 76-118 strain yielded its encoding gene (Appendix A) via nBLASTx (GenBank accession number: P08668.1). The DNA segment was inserted into the XbaI and KpnI sites of the pVAX1 vector and named pVAX-Gnc (Figure 1a). The recombinant plasmids were synthesized by Xi’an Branch of Wuhan Airco Biotechnology Co., Ltd. (Xi’an, China). The eukaryotic expression vectors were then synthesized with restriction enzymes. Gnc and LAMP/Gnc protein expression were confirmed by fluorescence confocal staining and EGFP observation. Sanger sequencing (Tsingke Biotech Co., Ltd., Xi’an, China) was used to validate the plasmid constructs. The recombinant plasmids, transformed into DH5α, were purified to eliminate endotoxins and adjusted to a concentration of 1 mg/mL in 1× phosphate-buffered saline (PBS). The plasmid pVAX-LAMP was obtained from our laboratory. The pVAX-LAMP/Gnc plasmid was cloned by inserting the Gnc sequence between the luminal and transmembrane/cytoplasmic LAMP domains at the XhoI and EcoRI sites at 114 bp at the 5′ end of the stop codon (Figure 1b,c).

### 2.4. Expression Identification of DNA Vaccines

For the expedient analysis of protein expression, Gnc in the form of a single or LAMP1 fusion was introduced into the vector pEGFP-N3, which were named pEGFP-Gnc and pEGFP-LAMP/Gnc, respectively (Figure 1d,e). EGFP refers to enhanced green fluorescent protein, GenBank ID: DD258990.1. Sanger sequencing was used to validate the plasmid constructs. The recombinant plasmids produced by transforming DH5α were purified to remove the endotoxin and then adjusted to a concentration of 1 mg/mL in 1× phosphate-buffered saline (PBS). Diamidino–phenyl–indole (DAPI, Solarbio, Beijing, China) and LysoTracker Red (Beyotime, Shanghai, China) were used for nuclear and lysosomal staining at room temperature. EGFP-tagged Gnc and LAMP/Gnc protein expression were also studied in HeLa cells. Positive signals for Gnc or LAMP/Gnc protein expression were observed using a fluorescence confocal microscope.

Fluorescent confocal staining was performed to assess the expression of Gnc and LAMP/Gnc proteins. HeLa cells were seeded in 6-well plates (1 × 10^6^ to 5 × 10^5^ cells per well) during their logarithmic growth phase and transfected with 4 μg of EGFP-labeled Gnc and LAMP/Gnc plasmid DNA using Lipofectamine™ 3000, following the manufacturer’s protocol. After 36 h of culture, suspensions were discarded and the cell slides were rinsed three times with cold PBS. Fixation was carried out using 4% paraformaldehyde for 15 min at room temperature, followed by additional rinses with PBS. Cells were then permeabilized with 0.5% Triton for 15 min at room temperature and blocked with a 3% bovine serum albumin (BSA) PBS solution for 30 min. After further rinsing with PBS, diamidino–phenyl–indole (DAPI) and LysoTracker Red (Beyotime) were used for nuclear and lysosomal staining, respectively. The expression of EGFP-tagged Gnc and LAMP/Gnc proteins in HeLa cells were observed, with positive signals detected using a fluorescence confocal microscope.

### 2.5. Physicochemical Property Analysis of DNA Vaccines

PeptideCutter is a tool designed to predict potential cleavage sites in a protein sequence that may be recognized by proteases or chemicals. It returns the queried sequence with possible cleavage sites marked and/or provides a table indicating the positions of these sites.

ProtParam is a resource that computes various physical and chemical parameters for a given protein sequence entered by the user. The parameters computed include molecular weight, theoretical isoelectric point, amino acid composition, atomic composition, extinction coefficient, estimated half-life, instability index, aliphatic index, and the grand average of hydropathicity (GRAVY). These parameters were analyzed for the Gnc protein.

The PSIPRED Protein Analysis Workbench is a well-known online service offering a variety of tools for protein prediction and annotation, primarily focusing on structural annotations. The secondary structure of the Gnc protein was evaluated using the PSIPRED 4.0 tool available on their server.

To predict the tertiary structure of multiepitope constructs, the I-TASSER web server was utilized. The tertiary structure of a protein is critical for its biological function. I-TASSER employs a hierarchical approach to predict protein structures and perform structure-based functional annotations. It constructs 3D atomic models through various stringing arrangements and iterative structural assembly simulations based on amino acid sequences.

The GalaxyRefine server [27] was employed to refine the predicted tertiary structures. This approach entails rebuilding side chains, carrying out side chain repacking, and relaxing the overall structure through molecular dynamics simulations, which generally enhance both the global and local quality of the structure.

Immunogenicity is determined solely using the peptide sequence and not its spatial structure. VaxiJen 2.0 [28] was used to predict the immunogenicity of the Gn, Gc, GP, and Gnc proteins. The GP (accession no: KT885048.1) from HTNV 76-118 was sourced from the NCBI GenBank.

The hydrophilicity of the Gnc protein was analyzed using an online tool from Novopro (https://www.novopro.cn/tools/protein-hydrophilicity-plot.html, (accessed on 8 December 2022)) [29].

### 2.6. Animals and Immunization

Eight-week-old female BALB/c mice, free from specific pathogens, were obtained from the Laboratory Animal Center of the University and randomly assigned into five groups (20 mice each). Groups A, B, C, and D were respectively injected at the base of the tail with 50 μg of four endotoxin-free DNA plasmids (pVAX-LAMP/Gnc, pVAX-Gnc, pVAX-LAMP, and pVAX). Group E received 20 μL of inactivated HTNV vaccine, following the manufacturer’s guidelines for the immunizing dose. As shown in Figure 2c, a total of three plasmid immunizations were injected on days 1, 21, and 42. The animals were well cared for, and all procedures followed animal experimentation practices. Two weeks after the third immunization, serum was collected from the mice for neutralization experiments, and then 5 mice were randomly selected for viral challenge. The remaining mice in each group were randomly sacrificed for the ELISPOT experiment three weeks after the third immunization.

### 2.7. Serum Neutralization Test

Vero E6 cells were cultured in 24-well plates using RPMI-1640 medium with 10% fetal calf serum and incubated at 37 °C for 18–24 h. Mouse sera were filtered through 0.22 μm filters, then serially diluted twofold in RPMI-1640 with 2% FCS starting at a 1:10 dilution and mixed with an equal volume of a 100× TCID50 suspension of HTNV (76–118 strain). After a 90 min incubation, this mixture was added to the cultured cells and incubated at 37 °C in a 5% CO_2_ environment for 9–11 days. The cells were lysed through three freeze–thaw cycles, and the HTNV antigen in the lysates was detected using a sandwich ELISA. The capture antibody was a monoclonal mouse anti-NP antibody (1A8), and the detection antibody was HRP-conjugated 1A8. Mouse monoclonal antibodies against Gn and Gc, along with Sp2/0 ascites, served as positive and negative controls, respectively. The absorbance was measured at 490 nm using an ELISA plate reader (Bio-Rad, Berkeley, CA, USA) and the neutralizing antibody titer was calculated using Karber’s method, defined as the highest serum dilution that inhibits HTNV infection in 50% of the cells.

### 2.8. Enzyme-Linked Immunospot Assay (ELISpot)

Interferon (IFN)-γ ELISpot reagents were sourced from BD Pharmingen (Franklin Lakes, NJ, USA) and evaluated according to the manufacturer’s instructions. Briefly, the ELISpots were coated with an IFN-specific capture antibody diluted in sterile PBS to 5 μg/mL overnight at 4 °C. Mice were sacrificed and their spleens were processed to obtain a mononuclear cell suspension. After washing and resuspending the spleen cells post-erythrocyte lysis, the plates were blocked with RPMI-1640 containing 10% fetal bovine serum for 2 h at room temperature. Each well received 1 × 10^6^ splenocytes, which were stimulated with synthetic GP peptide at a final concentration of 20 μg/mL. Pure RPMI medium served as a negative control, while Con A (5 μg/mL) was used as a positive control. The plates were incubated at 37 °C for 24 h in a 5% CO_2_ incubator. After incubation, the plates were washed with ddH2O and PBST, followed by incubation with 2 μg/mL of biotinylated rat anti-mouse IFN-γ antibody for 2 h at room temperature. After washing with PBST, streptavidin-HRP was added at a 1:100 dilution for 1 h. The HRP substrate 3-amino-9-ethylcarbazole (AEC, BD Pharmingen) was then added, and the reaction was stopped with water washes. The IFN-γ spots were counted using the AID ELISpot Reader Classic-ELR06 (AID, Strassberg, Germany) after the plates were air-dried. Results are expressed as the mean number of spot-forming cells (SFCs) per 10^6^ splenocytes.

### 2.9. Viral Challenge of Mice

Twelve days after the final immunization, five mice from each group were intraperitoneally challenged with 1 × 10^5^ plaque-forming units (PFUs) of the HTNV 76–118 strain. This challenge occurred on day 192, and the mice were sacrificed on day 195. Major organs (heart, liver, spleen, lungs, kidneys, and brain) were harvested, and the viral load in these organs was assessed via quantitative reverse-transcription (qRT)-PCR to evaluate vaccine protective efficacy. The target sequence of the HTNV S segment was amplified using the primers HTNV S-forward (5′-GAT CAG TCA CAG TCT AGT CA-3′) and HTNV S-reverse (5′-TGA TTC TTC CAC CAT TTT GT-3′), with glyceraldehyde-3-phosphate dehydrogenase (GAPDH) serving as the control. Results were recorded as quantitative circulation (Cq) and quantified using the following formula: Cq(experiment) − Cq(control). The inactivated vaccine was used as a responsive control due to its protective efficacy against HTNV infection. All experiments were conducted under BSL-3 conditions, following the International Laboratory Biosafety Manual guidelines.

### 2.10. Immune Simulation after Three Doses of Vaccines

Immune stimulation was conducted using the C-ImmSim server [30,31,32], which predicts alterations in T and B lymphocyte expression and cytokine levels with vaccination. The parameters included a random seed, default simulation volume, and the most prevalent HLA genotype in the Asian population according to the Allele Frequency website, specifically HLA-A0101, HLA-A0201, HLA-B0702, HLA-B0704, HLA-DRB1-0101, and HLA-DRB1-0102. The simulation consisted of 800 steps, with 3 vaccine doses administered on days 0, 21, and 42 [33].

### 2.11. Hematoxylin and Eosin (H&E) Staining

Two weeks post the third immunization, major organs and skin from the injection sites were collected. After being fixed in 4% formaldehyde, the tissues were embedded in paraffin, sliced, and stained with hematoxylin and eosin (H&E) for histopathological examination under a microscope.

## 3. Results

### 3.1. Rationale Design, Physicochemical Properties, and Structures of the Antigen Gnc

According to the IFN-γ responses of peripheral blood mononuclear cells (PBMCs) [20] and splenocytes from HFRS-immunized BALB/c mice immunized with the inactivated bivalent HFRS vaccine (HANPUWEI^®^) [21,22,23], a heatmap was drawn to visualize the effects of favorable stimuli on HTNV Gn and Gc (Figure 2a). Sequentially, a recombinant antigen should construct a collection of 51 dominant Th epitopes of both the Chinese Han population and BALB/c mice on the HTNV GP. The novel protein Gnc consists of 459 amino acids, and 10 antigenic peptides (aa19-40, aa145-215, aa236-271, aa285-306, aa369-383, aa418-509, aa551-572, aa711-761, aa775-852, and aa957-985 of HTNV GP) are concatenated with the preferred lysosomal enzyme-hydrolyzing amino acid KK [34].

PeptideCutter (http://web.ExPASy.org/peptide_cutter/, (accessed on 17 January 2024)) was used to predict the restriction sites of Gnc amino acid sequences in 23 enzymes. Among them, for the three enzymes LysC, LysN, and trypsin, the toolkit predicted 38, 38, and 49 restriction sites, respectively, which were close to the hydrolysis sites of Gnc.

The physicochemical characteristics and structures of vaccine molecules significantly influence their immunogenicity [35]. The Gnc protein’s properties were analyzed using various tools. The ProtParam server indicated that the vaccine construct had a molecular weight of 51.342 kDa [36]. Vaccines typically have a molecular weight under 110 kDa, which is suitable for purification processes [37]. A theoretical isoelectric point (pI) of 8.60 was calculated, assisting in tracking the vaccine on a 2D gel. The protein contained 39 negatively charged (Asp + Glu) and 50 positively charged (Arg + Lys) residues, with charged residues and Cys making up 5.9%. The extinction coefficient at 280 nm in water was measured at 63,395, indicating that each pair of Cys residues was a cysteine. The estimated half-life was 30 h for mammalian reticulocytes (in vitro), over 20 h for yeast (in vivo), and over 10 h for *Escherichia coli* (in vivo). The instability index of Gnc was found to be 32.81, suggesting the antigen’s relative stability. The ProtParam server also calculated an aliphatic index of 90.81 and a grand average of hydropathicity of 0.136, classifying Gnc as hydrophobic.

The secondary structure of the Gnc protein was predicted by the PSIPRED Workbench (http://bioinf.cs.ucl.ac.uk/psipred, (accessed on 3 October 2022)) [38,39], which analyzes segments of α-helices, β-sheets, and random coils, and visualizes them in different colors (Figure 2b).

I-TASSER (https://zhanggroup.org/I-TASSER/, (accessed on 14 October 2022)) [40,41] simulated multiple structural conformations of Gnc molecules and provided the top five most likely three-dimensional structures. Appendix A provides molecular models for five PDB formats. Molecular dynamics simulations with five model structures were subsequently performed and improved using the GalaxyRefine server (https://galaxy.seoklab.org/index.html, (accessed on 17 October 2022)) [42]. High-accuracy models of the frame structure were obtained by refining the local region with structural errors through a cyclic modeling approach. For a prediction model, the algorithm provided five optimized models in PDB format. Appendix A provides all the optimization results for the five models in I-TASSER. Figure 2c shows the most likely model predicted by I-TASSER and the five optimized models. Table 1 shows the structural information of Model 1 and the five optimized models. Improvement in model quality by refinement was measured in terms of improvement in GDT-HA [43], Cα-RMSD (RMS-CA), the SphereGrinder score (SphGr) [44], and the Molprobity Score (MolPrb) [45].

According to the predictions of VaxiJen (http://www.ddg-pharmfac.net/vaxijen/VaxiJen/VaxiJen.html, (accessed on 30 September 2023)) [46], the immunogenicity scores of all four proteins were above the threshold. Gc had the strongest immunogenicity, followed by Gnc, with Gn having the lowest immunogenicity. Specific overall predictions for the protective antigen scores are provided in Appendix A.

The Gnc protein tends to be hydrophobic from aa1–aa270 and hydrophilic after aa270. The hydrophilicity diagram is presented in Appendix A.

According to the above analysis, the Gnc molecule is a hydrophilic, immunogenic, and stable molecule, suggesting that the recombinant protein is appropriate for use as a vaccine antigen. Based on the secondary and tertiary structural simulations, the probable modes of action of the recombinant protein molecules were preliminarily recognized.

### 3.2. Synthesis and Expression of Recombinant DNA Plasmids

Gnc and LAMP/Gnc protein expression was confirmed by fluorescence confocal staining and EGFP observation. As shown in Figure 1f, the expression of the vectors pEGFP-Gnc and pEGFP-LAMP/Gnc in HeLa cells was observed. EGFP fluorescence suggested that the target recombinant protein Gnc or LAMP/Gnc was successfully expressed in cells; DAPI fluorescence showed the location of the nucleus; RFP represented the lysosomal membrane probe; and its fluorescence showed the lysosomal location and reflected the entry of the recombinant LAMP/Gnc into lysosomes. Both plasmids could be expressed normally in eukaryotic HeLa cells.

Among them, the brightness of the Gnc group was greater than that of the LAMP/Gnc group, which may be due to the influence of fusion gene length and LAMP1 glycosylation on the expression of the LAMP/Gnc protein. However, at the site of expression, LAMP/Gnc-EGFP was partially present in lysosomes, suggesting that the vector vaccine fused with LAMP1 could be effectively trafficked into lysosomes.

### 3.3. Protective Efficacy of Tandem Epitope Vaccines against HTNV Infection in BALB/c Mice

The effect of the vaccine on the virus in different organs varies [47]. RT-qPCR is the gold standard for viral load detection [48]. The viral loads of major organs in the body were measured 3 days after challenge in the mice. As shown in Figure 3, the viral load in the main organs of mice in the experimental group decreased compared with that in the inactivated vaccine group, and the decrease was greater in the heart, liver, and spleen; moreover, the protective efficacy was similar in the kidney, lung, and brain.

### 3.4. LAMP Targeting Enhances the Cellular Response to Gnc-Encoding DNA Vaccines with Incompetent Neutralization

Neutralizing antibodies are considered to be key tools for virus clearance [49]. However, no neutralizing activity was observed for Gnc-related vaccine candidates. The neutralizing antibody titers of pVAX-Gnc and pVAX-LAMP/Gnc were comparable to those of the control. The neutralizing antibody titer of 1:49.5 was detected for the inactivated vaccine, which coincides with previous observations [ [50]].

The specific cellular immune responses of mouse spleen cells were detected using an enzyme-linked immunospot (ELISpot) assay. The animals were well cared for, and all procedures followed animal experimentation practices. The frequency of IFN-γ secretion by T-cells was measured using an ELISpot assay to evaluate cellular immune responses to vector vaccines. As shown in Figure 4, 47 highly reactive epitopes were used to stimulate mouse splenocytes after immunization in the three groups. Among them, the above 47 epitopes could induce cellular immune responses in all three groups of mice, confirming the effectiveness of the epitopes. At the same time, mice in the pVAX-LAMP/Gnc group had greater splenocyte-specific IFN-γ secretion, which demonstrated that the specific cellular immune response of mice in this group was significantly increased at the single-cell level, while splenocyte IFN-γ secretion was slightly increased in mice in the inactivated vaccine group compared with that in the pVAX-Gnc group, suggesting that targeted presentation of LAMP molecules enhanced the effect of tandem epitope vaccines. The P value of two-way ANOVA for three groups was 0.002.

### 3.5. Determination of the Preliminary Safety of Candidate DNA Vaccines Encoding Gnc Using Histopathological Analysis

No significant changes were detected in the major organs of the mice in the pVAX-LAMP/Gnc, pVAX-Gnc, HANPUWEI^®^, and normal control groups (Figure 5). No pathological alterations were observed despite the greater subcutaneous immune infiltration in the vaccinated groups than in the normal control group according to the H&E staining results. The same pattern was observed in the control pVAX and pVAX-LAMP groups. These findings offer preliminary evidence for safety related to immunization with pVAX-LAMP/Gnc and pVAX-Gnc plasmids.

### 3.6. LAMP1 Strategy Enhanced the Immune Response of T-Cells in the Population

C-IMMSIM (https://kraken.iac.rm.cnr.it/C-IMMSIM/index.php, (accessed on 10 June 2023)) was used to analyze the immune effects of two DNA vaccine candidates in the major Asian population. The results (Figure 6 and Appendix A) showed that both vaccines were effective in activating cellular and humoral immune responses in humans after three doses. There was little difference in the level of antibody responses stimulated by the two vaccines. However, the T-cell response differed; the number of T- and Th cells stimulated by the pVAX-LAMP/Gnc vaccine was greater than that stimulated by the pVAX-Gnc vaccine. Curiously, the pVAX-LAMP/Gnc vaccine had similar IFN-γ levels after the third immunization, but the pVAX-Gnc vaccine had a precipitous decrease in IFN-γ levels after the third immunization.

As shown in Figure 6e,f, the number of Th memory cells produced by the pVAX-LAMP/Gnc vaccine after three immunizations was significantly greater than that produced by the pVAX-Gnc vaccine, suggesting that the LAMP strategy might enhance the long-term immune response and potential protective effect.

IL-4 has long been known to promote effective B-cell survival and isotype class switching [51]. However, as shown in Figure 6e,f, there was less IL-4 production after the injection of both recombinant vaccines, which may be the reason for the low levels of neutralizing antibodies in the trial results. On the other hand, IFN-γ was significantly greater after three immunizations in the pVAX-LAMP/Gnc group than in the pVAX-Gnc group, which was consistent with the results of the ELISpot assays. The above phenomenon also suggested that the production of IFN-γ may decrease due to immune tolerance after repeated immunization with the pVAX-Gnc vaccine, and the LAMP strategy will be an ideal method to overcome this tolerance.

## 4. Discussion

Vaccines are arguably the most successful biomedical advances in preventing disease. Epitope-based vaccines could confer advantages over traditional vaccines as they are smaller and may elicit a focused immunoresponse with neutralizing activity toward antigens [52,53,54,55]. In this study, we combined highly immunoreactive epitopes in human-derived and murine-derived cells [56,57] to construct the tandem recombinant DNA vaccine pVAX-Gnc and optimized it for LAMP1 targeting. The above vaccine vectors were shown to be normally expressed in eukaryotic cells by fluorescence confocal staining and eGFP observation, and after LAMP1 optimization, the distribution of the target protein in lysosomes significantly increased. By analyzing the physicochemical properties and structure of the Gnc protein in silico, we obtained a preliminary understanding of the applicability of the target protein as a vaccine. After BALB/c mice were immunized three times, the ELISpot test and challenge test were performed. The results showed that the epitope vaccine could effectively activate a specific antiviral immune response, and the strategy of targeting lysosomes with LAMP1 could significantly improve the specific cellular immune response and challenge protection efficacy in mice. Hantavirus glycoprotein is an immune target molecule. Two nucleic acid vaccines have been constructed and improved by rational design of immune reactive epitopes on the glycoprotein. The vaccine has been validated in eukaryotic cells and mice, preliminarily demonstrating its ability to express and produce immune-protective effects within cells. Through bioinformatics analysis, it has been preliminarily proven that vaccines have strong protective effects on populations worldwide. Both vaccines were predicted to induce strong IFN-γ secretion during population immunity. The study provided the basis for the development of antiviral therapy with epitope vaccines.

Peptide-/epitope-based vaccine strategies have been widely adopted in the prevention and treatment of infectious diseases, tumors, and other diseases [58,59,60]. Currently, increasing studies integrate bioinformatics with vaccine development. The pathogen-specific epitopes are usually obtained using computer analysis [61,62,63], and then the efficacy would be evaluated by immunological tests after vaccination [59,64,65]. However, a persistent challenge for peptide vaccines is the comparatively weaker overall immune response in comparison to traditional inactivated or attenuated candidates [66]. Therefore, our study selected highly cellular immunoreactive epitopes that have been demonstrated in humans and mice species, which avoided inaccuracies in immune targets. At the same time, we verified the epitopes by in vitro testing of vaccines for immunization, which established a link between the results of in vitro tests and the immune response in vivo, and then evaluated the protective efficacy in immunized mice by evaluating the effectiveness of the epitopes in the organism again.

The cytoplasmic tail of the LAMP molecule is targeted and directionally binds the subcellular structure, consisting of endosomes/lysosomes, through its membrane-penetrating cytoplasmic region, and flips into it [67]. Recent studies have shown that DNA vaccines targeting the LAMP1 sequence induce strong cellular immunity and initiate a strong Th1 response [21,22,23,68,69]. The Gnc gene was inserted between the LAMP molecule encoding the endolysosomal portion and the cytoplasmic tail gene to construct a chimera. The experimental results showed that the specific cellular immune response of mice in the LAMP/Gnc group was significantly greater than that in the other groups, suggesting that pVAX-LAMP/Gnc induced strong cellular immunity. This strong immune response may be due to the transport of LAMP/Gnc chimera into the MHC-II compartment, which achieves the conversion of proteins expressed by DNA vaccines from the endogenous MHC-I processing pathway to the exogenous MHC-II processing pathway, resulting in an increased efficiency of antigen processing and epitope binding to MHC-II.

The epitope specificity of T-cells is mediated by the T-cell receptor (TCR), which binds peptides present in the “peptide-binding groove” of MHC-I or MHC-II on antigen-presenting cells (APCs) [70,71]. However, MHC polymorphisms result in epitopes with different affinities on different MHCs [72,73]. Therefore, determining how to make the vaccine applicable to a wider range of MHC genotypes, populations, and regions is not only a deficiency of this study but also an urgent problem in epitope vaccines that has long needed to be solved [52,74]. On the other hand, tandem epitope vaccines and targeted lysosomal tandem epitope vaccines have performed excellently in cellular immune evaluation; however, the epitopes involved are limited to cellular immunoreactive epitopes and are deficient in humoral immune epitopes, suggesting that the antiviral immune response in vivo in the absence of neutralizing antibodies may be established by protective T-cells. The existing research strategy is not sufficient to identify the source of the protective cellular immune response. Of course, this is only a great attempt, and subsequently, the addition of humoral immunoreactive epitopes should be considered, as well as codon optimization and promoter selection [75,76], for the next version of the tandem epitope vaccine.

Our study was based on the principle of reverse vaccinology. The epitope vaccine pVAX-Gnc was synthesized and then optimized with LAMP1 to achieve antigen targeting to lysosomes. The efficacy of the vaccine suggested that the Gnc protein can be used as a candidate target against HTNV, and the application of Gnc to DNA vaccines effectively solved the shortcomings of peptide vaccines. Strategies to target antigen presentation to lysosomes have been shown to markedly enhance immune responses in mice. The trains of thought in epitope screening and vaccine construction involved in this study can also be applied in the prevention and control of other pathogens, such as SARS-CoV-2 and monkeypox virus, by adopting bioinformatics approaches to appraise epitopes [77,78,79,80]. This study is a grand practice from the construction of vaccines to the evaluation of protective efficacy, combining in vitro test results with in vivo studies, and provides a novel approach to designing vaccines against multiple pathogens.

## Figures and Tables

**Figure 1 vaccines-12-00928-f001:**
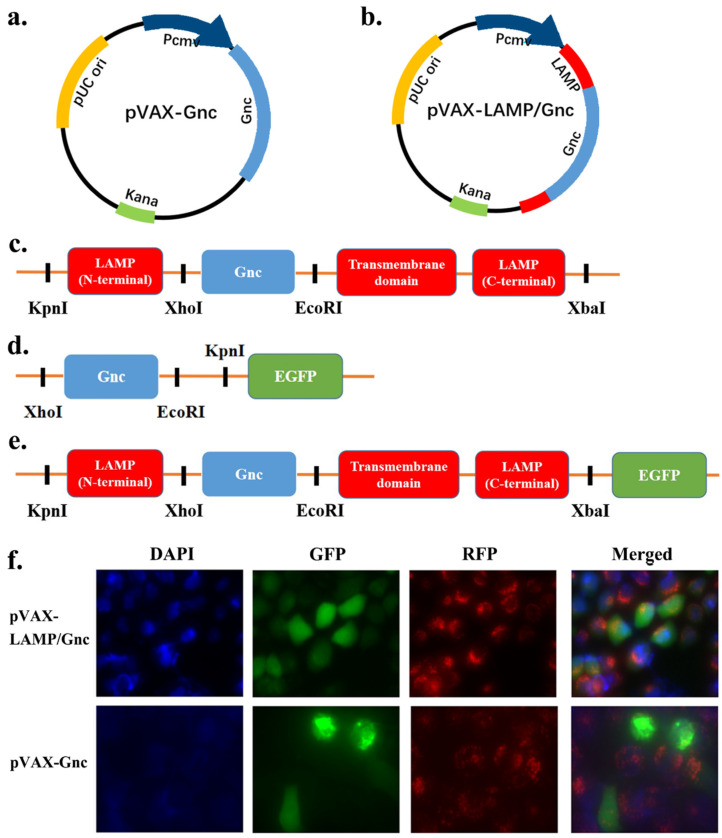
Construction of candidate DNA vaccines and eukaryotic expression vectors and protein identification in HeLa cell line. (**a**,**b**) Schematic diagram of the pVAX-Gnc and pVAX-LAMP/Gnc vaccine structures; (**c**) schematic representation of Gnc gene insertion into the LAMP1 molecule; (**d,e**) schematic representation of the structures of the pEGFP-Gnc and pEGFP-LAMP/Gnc eukaryotic expression vectors; (**f**) identification of Gnc and LAMP/Gnc by immunofluorescence staining and confocal microscopy. Blue: DAPI, labeled cell nucleus; green: GFP, the target recombinant protein Gnc or LAMP/Gnc; red: RFP, lysosomal probe.

**Figure 2 vaccines-12-00928-f002:**
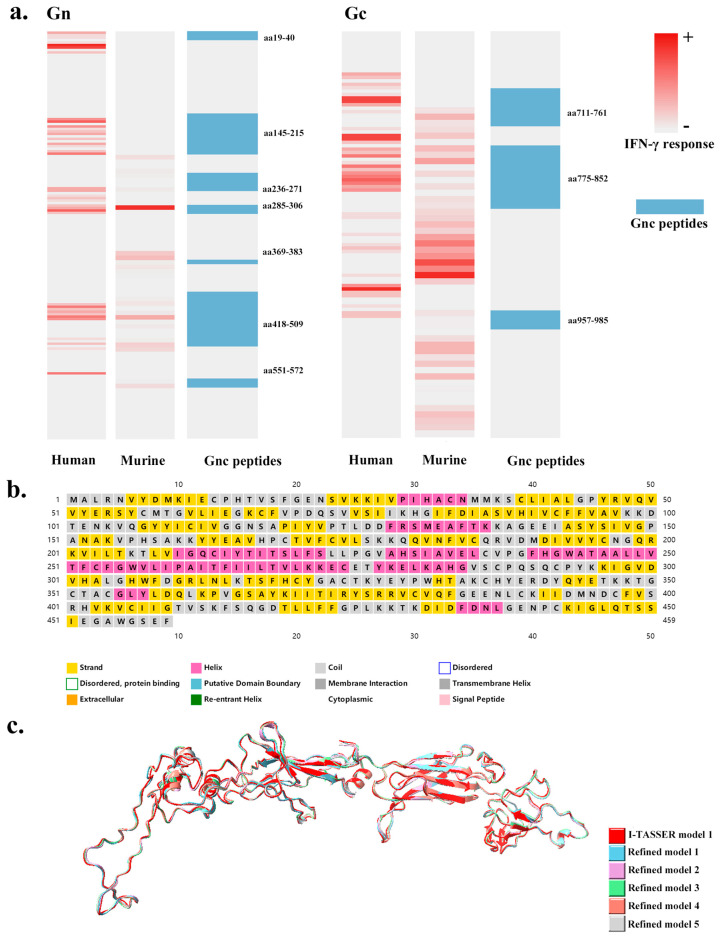
The rational design, physicochemical properties, and structures of the recombinant antigen Gnc. (**a**) The amount of IFN-γ response stimulated by the Gn and Gc epitopes in human (HFRS patients) and murine lymphocytes; the darker the red color is, the stronger the IFN-γ response; the selection of Gnc peptides; the blue part shows the selected GP peptides; (**b**) the figure shows the predicted secondary structure for each amino acid site; yellow for α-helix, pink for β-coil, gray for random coil; (**c**) the most likely model predicted by I-TASSER and five optimized models.

**Figure 3 vaccines-12-00928-f003:**
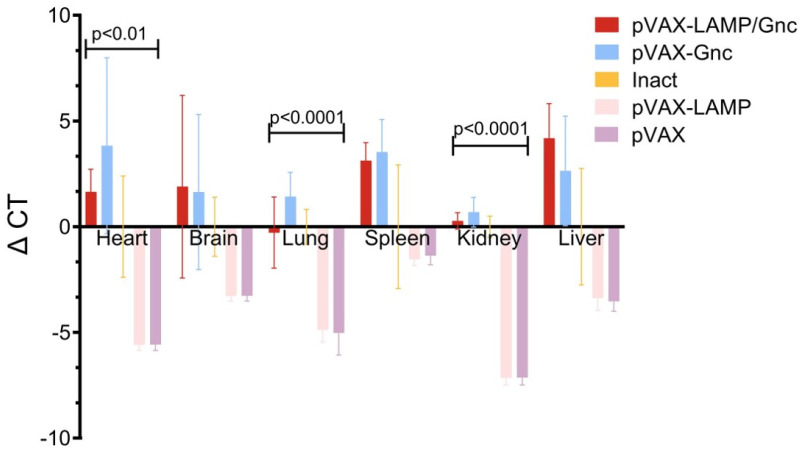
Detection of viral load in major organs after the challenge test. The ΔCT of the inactivated vaccine group was used as the baseline. The values displayed in each group are the ΔCT values minus the average of the inactivated group.

**Figure 4 vaccines-12-00928-f004:**
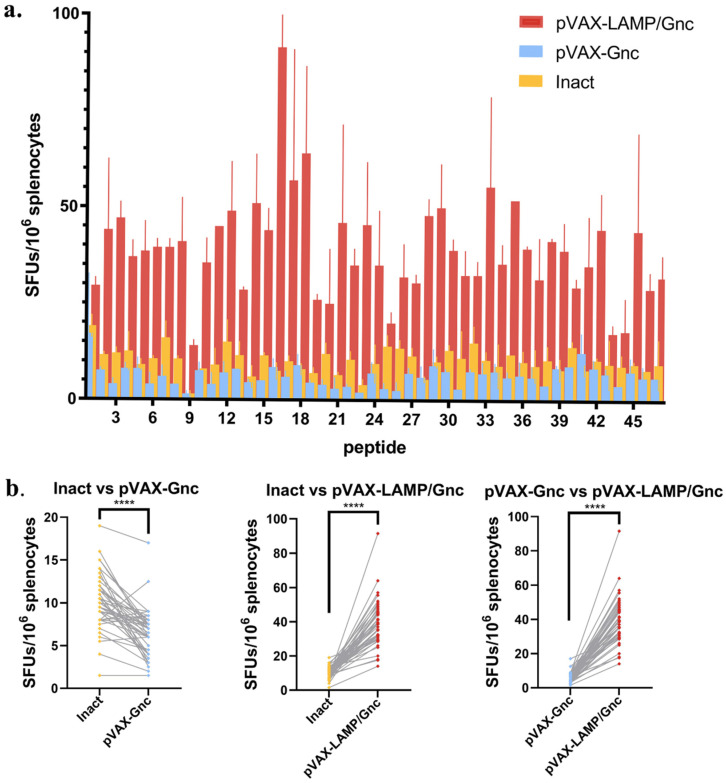
The DNA vaccines elicited IFN-γ secretion in BALB/c mice. (**a**) Splenocyte IFN-γ secretion in response to 15-mer peptides in the three vaccination groups revealed that chimeric pVAX-LAMP/Gnc significantly enhanced the IFN-γ secretion response of T-cells. (**b**) The paired *t*-test pattern for pairwise comparisons. **** *p* < 0.0001.

**Figure 5 vaccines-12-00928-f005:**
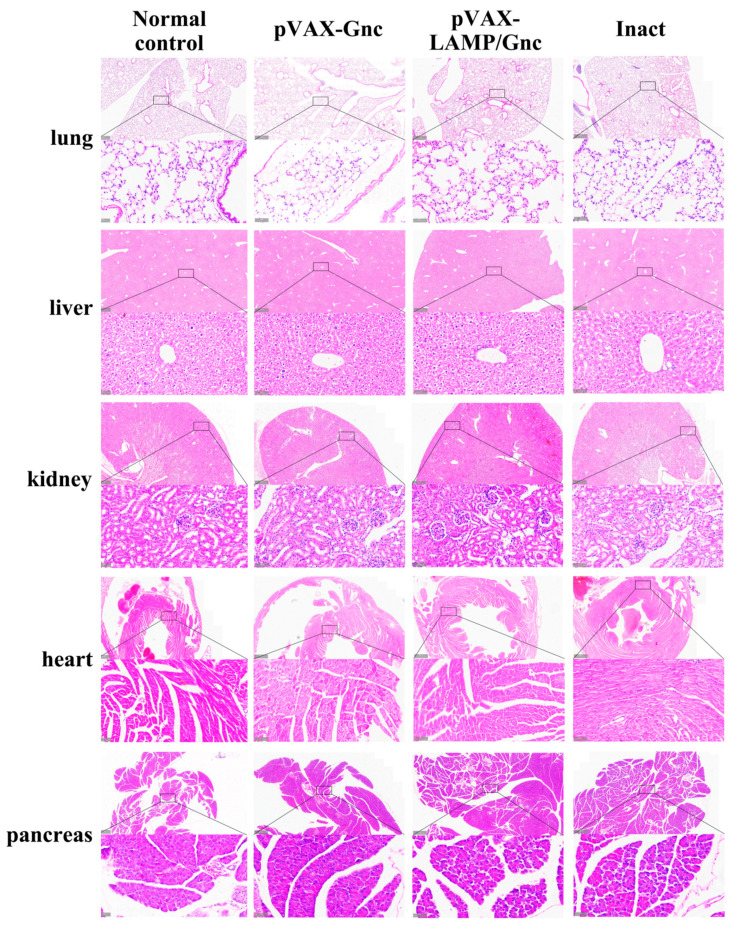
Histological analysis for subacute toxicity of pVAX-Gnc and pVAX-LAMP/Gnc. H&E staining indicated that no significant differences existed between the chimeric DNA-innoculated and normal mice.

**Figure 6 vaccines-12-00928-f006:**
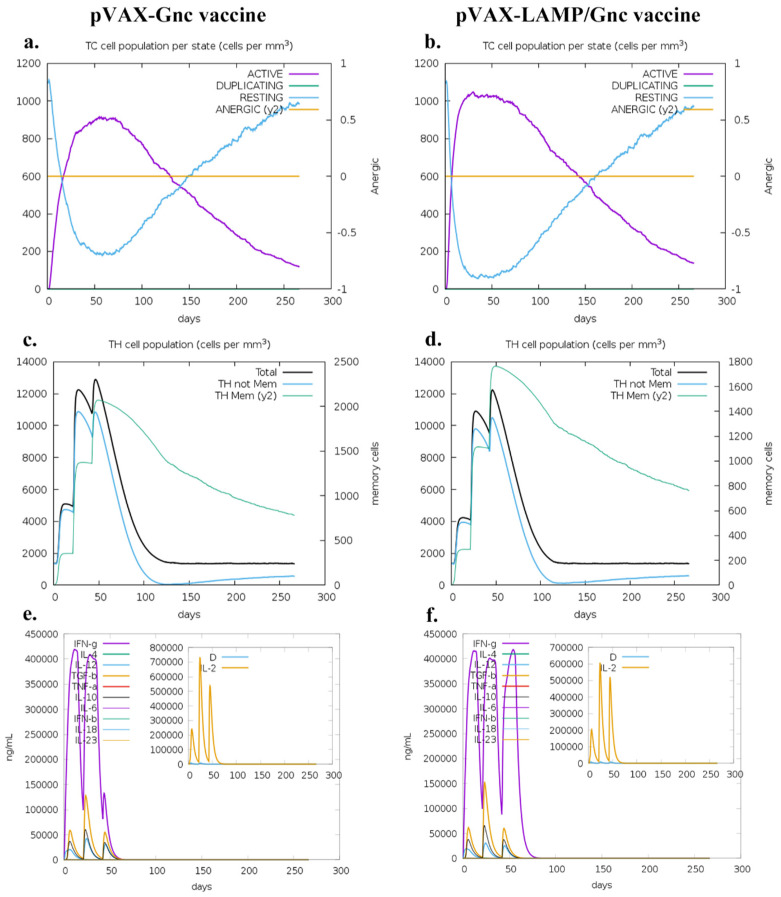
Prospective responses of major populations to Gnc candidate vaccines. (**a**,**b**): CD8 T-cytotoxic lymphocyte count per entity-state; (**c**,**d**) CD4 T-helper lymphocyte count; the plot shows total and memory counts; (**e**,**f**) cytokines, concentrations of cytokines and interleukins; D in the inset plot is the danger signal.

**Table 1 vaccines-12-00928-t001:** The structural information of Model 1 and five optimized models.

Model	GDT-HA	RMSD	MolProbity	Clash Score	Poor Rotamers	Rama Favored
Model 1	1	0	3.611	20.1	17.7	67.2
Refined 1	0.8981	0.525	2.642	31.4	1	84.9
Refined 2	0.9112	0.517	2.624	32.9	1	86.9
Refined 3	0.9031	0.529	2.685	31.7	1.2	86.7
Refined 4	0.9009	0.528	2.715	31.1	1.2	84.7
Refined 5	0.8998	0.532	2.575	28.9	0.7	86.7

## Data Availability

Data are contained within the article or Appendix A.

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
