# Peer review of "DNA Vaccines Encoding HTNV GP-Derived Th Epitopes Benefited from a LAMP-Targeting Strategy and Established Cellular Immunoprotection"

_vaccines, 2024, doi:10.3390/vaccines12080928_

Round 1

Reviewer 1 Report

Comments and Suggestions for Authors

The manuscript entitled “DNA vaccines encoding HTNV GP-derived Th epitopes benefited from a LAMP-targeting strategy and established cellular 3 immunoprotection” presents the work on vaccines development. Arguably the most successful biomedical advances in preventing diseases is vaccine. Moreover, the study explores the use of epitope-based vaccines, this particular study is important because it uses targeted vaccine approach. The use of subunits could elicit specialized antibodies there by reducing the reactivity of the body to other activities that are not needed.  The text and the logic flow well with the introductions, methods and results. The discussions are well written and discuss the results well. However, the manuscript will benefit from reading if the authors could address few minor comments bellow.

Broder comment

Since the authors are working with the protein expression, plasmids ect, it will be good to show the length of the hantavirus genome in the introduction including the length of the aa coded by these segments.

Minor comments

What is ChinaPeptides? Is it online database, it will improve reading if the authors could add the citation and complete the sentence by adding that it is an online database.

Lines 160-162: it is not clear whether the authors vaccinated each group with the four types of plasmids they created.

In the manuscript it is not clearly stated what exactly the authors put in the plasmids. Did they put only for example only immunogenic part for the Gn etc or they put the full coding sequences in the plasmids they constructed.

Line 169: the authors may wish to avoid using killed but more partial words like were sacrificed.

Lines 234-238: what is the purpose of these sentences here? Looks like they are coming from somewhere, on this paragraph they don’t make sense.

Line 239-241: also looks like the sentences are not on the designated place?

Figure 1 C can be improved.

Line 308, why the yellow come twice. I think it is typo.

Lines 313-326: can be summarized, some information can be taken to the materials and methods section.

Author Response

Dear referee,

I would like to express my heartfelt gratitude for your recognition and approval of our work. Your feedback and suggestions have greatly improved the quality and clarity of the manuscript. We have taken your comments into careful consideration and have made the necessary revisions to improve the overall quality of our manuscript.

Please refer to the attached file for details on addressing all the comments.

Reviewer 2 Report

Comments and Suggestions for Authors

In the manuscript “DNA vaccines encoding HTNV GP-derived Th epitopes benefited from a LAMP-targeting strategy and established cellular 3 immunoprotection”, Dongbo Jiang et al., describe LAMP based DNA vaccines development. The authors using reverse vaccinology shows that the LAMP—based vaccine can improve the protection compared to the traditional vaccine methods. This part of the study is important and it is a piece of work which adds knowledge to the field of vaccine development. Most importantly, the study approach can be applied to other viruses too. The text of the manuscript flows well with the idea presented. The list of references is on the topic. In the manuscript there few comments that need to be addressed before the manuscript is published.

Minor comments:

In the manuscript the authors use the old name Hantaan virus: they would like to check the ICTV for the updated name of Hantaan virus.

The authors would like to check all online database if they are all well cited.

From line 160 the vaccinated groups are not well explained. Did each group receive independent plasmid or all plasmids where combined.

How did the authors identify the genes they inserted in the plasmids. Is it through in silico analysis with through HLA or MHC.

Materials 6, table 6 the 10 amino acids, some are too long. It will be good to add a text to the manuscript to justify this.

Some materials in the results section should be taken to materials and methods section. For example, the text between Lines 313 and 326.

The author would like to go through the text of the manuscript to remove some repetition.

Figure 1 A the words Gnc peptides on the right-side bellow IFN respond what does it describe.

The authors should carefully go through the manuscript to remove the typos for example yellow word is coming twice.

The authors could consider adding a table in the results of immunogenic epitopes they used for Gn Gc before they combined them.

In figure 1 B there is no blue color.

Did the authors use any linkers in the polypeptide they created figure 1B. if yes, this is not explained in the manuscript

Line 372-375: are these 47 epitopes used different from the ones used to create the LAMP Gnc.

It will be very interesting to see the three-dimensional representation of the TLR4–peptide docked complex. How figure 1C bind with TLR4.

Author Response

(The authors gave the same response as above.)

Reviewer 3 Report

Comments and Suggestions for Authors

This manuscript presents the development and evaluation of DNA vaccines encoding Hantaan virus (HTNV) glycoprotein (GP)-derived Th epitopes, which were then optimized with a lysosomal targeting membrane protein (LAMP).

Below are some suggestions and comments.

 1. Line 56-62, which could be redundant, has nothing to do with vaccine design.

 2. Protective T-cell epitopes and LAMP1-targeting practices are fundamental in this manuscript. It lacks a sufficient introduction.

 3.  Multiple online tools were utilized, it would be better to provide comprehensive analysis instruction.

 4.  Line 285, algorithm detail shall be briefly mentioned as it may affect the model results.

 5. Line 365, the cited paper is still under review. it’s not appropriate to count it as a reference.

 6. A comparison to the traditional inactivated and attenuated vaccine should be provided to demonstrate the advantages of the LAMP-targeting vaccine.

 7. More experiments such as in vivo T cell phenotyping utilizing flow cytometry should be designed and performed to prove the enhanced immunity.

Author Response

(The authors gave the same response as above.)

Round 2

Reviewer 3 Report

Comments and Suggestions for Authors

The author addressed most comments appropriately.